# Quantum-limited stochastic optical neural networks operating at a few quanta per activation

Shi-Yuan Ma [1] ✉, Tianyu Wang [1], Jérémie Laydevant[1,2], Logan G. Wright [1,3,5] & Peter L. McMahon [1,4] ✉

Energy efficiency in computation is ultimately limited by noise, with quantum limits setting the fundamental noise floor. Analog physical neural networks hold promise for improved energy efficiency compared to digital electronic neural networks. However, they are typically operated in a relatively high-power regime so that the signal-to-noise ratio (SNR) is large (>10), and the noise can be treated as a perturbation. We study optical neural networks where all layers except the last are operated in the limit that each neuron can be activated by just a single photon, and as a result the noise on neuron activations is no longer merely perturbative. We show that by using a physics-based probabilistic model of the neuron activations in training, it is possible to perform accurate machine-learning inference in spite of the extremely high shot noise (SNR ~ 1). We experimentally demonstrated MNIST handwritten-digit classification with a test accuracy of 98% using an optical neural network with a hidden layer operating in the single-photon regime; the optical energy used to perform the classification corresponds to just 0.038 photons per multiply-accumulate (MAC) operation. Our physics-aware stochastic training approach might also prove useful with non-optical ultra-low-power hardware.

The development and widespread use of very large neural networks for artificial intelligence[1,2] has motivated the exploration of alternative computing paradigms—including analog processing—in the hope of improving both energy efficiency and speed[3,4]. Photonic implementations of neural networks using analog optical systems have experienced a resurgence of interest over the past several years[5-15]. However, analog processors—including those constructed using optics—inevitably have noise and typically also suffer from imperfect calibration and drift. These imperfections can result in degraded accuracy for neural-network inference performed using them[5,16-18]. To mitigate the impact of noise, noise-aware training schemes have been developed[19-25]. These schemes treat the noise as a relatively small perturbation to an otherwise deterministic computation, either by explicitly modeling the noise as the addition of random variables to the processor's output or by modeling the processor as having finite bit precision. Recent demonstrations of ultra-low optical energy usage in optical neural networks (ONNs)[12,15] have used merely hundreds to thousands of photons (SNR ≲ $10^2$) to represent the neuron pre-activation signal prior to photodetection (or equivalently, <1 photon per MAC). However, they were still in the regime where the noise is a small perturbation. More typically, millions of photons per activation are used to achieve reliable, accurate results[10,11,15,26]. In this paper we address the following question: what happens if we use such weak optical signals in a ONN that each photodetector in a neural-network

[1]School of Applied and Engineering Physics, Cornell University, Ithaca, NY, USA. [2]USRA Research Institute for Advanced Computer Science, Mountain View, CA, USA. [3]NTT Physics and Informatics Laboratories, NTT Research, Inc., Sunnyvale, CA, USA. [4]Kavli Institute at Cornell for Nanoscale Science, Cornell University, Ithaca, NY, USA. [5]Present address: Department of Applied Physics, Yale University, New Haven, CT, USA. ✉e-mail: sm2725@cornell.edu; pmcmahon@cornell.edu

layer receives at most just one, or perhaps two or three, photons? As we will explain, in this scenario, the training methods that treat the photodetector outputs as deterministic values with a small amount of noise added to them, such as used in refs. [12,15], would fail to achieve high machine-learning accuracy, so a new approach is needed.

Physical systems are subject to various sources of noise. While some noise can be reduced through improvements to the hardware, some noise is fundamentally unavoidable, especially when the system is operated with very little power—which is an engineering goal for neural-network processors. Shot noise is a fundamental noise that arises from the quantized, i.e., discrete, nature of information carriers: the discreteness of energy in the case of photons in optics, and of discreteness of charge in the case of electrons in electronics[27]. A shot-noise-limited measurement of a signal encoded with an average of $N_p$ photons (quanta) will have an SNR that scales as $\sqrt{N_p}$[28]. The *shot-noise limit*, which is sometimes also referred to as the *standard quantum limit*[29], can be evaded if, instead of encoding the signal in a thermal or coherent state of light, a quantum state—such as an intensity-squeezed state or a Fock state—is used. In this paper we consider only the case of *classical* states of light for which shot noise is present and the shot-noise limit applies. To achieve a suitably high SNR, ONNs typically use a large number of quanta for each detected signal. In situations where the optical signal is limited to just a few photons, photodetectors measure and can count individual quanta. Single-photon detectors (SPDs) are highly sensitive detectors that— in the typical *click detector* setting—report, with high fidelity, the absence of a photon (*no click*) or presence of one or more photons (*click*) during a given measurement period[30]. In the shot-noise-dominated regime of an optical signal with an average photon number of about 1 impinging on an SPD, the measurement outcome will be highly stochastic, resulting in a very low SNR (of about 1). Again, this is under the assumption that the optical signal is encoded in an optical state that is subject to the shot-noise limit—which is the case for classical states of light. Conventional noise-aware-training algorithms are not able to achieve high accuracy with this level of noise. Is it possible to operate ONNs in this very stochastic regime and still achieve high accuracy in deterministic classification tasks? The answer is *yes*, and in this work, we will show how.

The key idea in our work is that incorporating a physics-based, probabilistic model of the highly stochastic photodetector outputs into the training algorithm can result in high-accuracy, deterministic inference with the ONN hardware. When ONNs are operated in the approximately-1-photon-per-neuron-activation regime and the detectors are SPDs, it is natural to consider the neurons as binary stochastic neurons: the output of an SPD is binary (*click* or *no click*) and fundamentally stochastic. Instead of trying to train the ONN as a deterministic neural network that has very poor numerical precision, one can instead train it as a binary stochastic neural network, adapting some of the methods from the last decade of machine-learning research on stochastic neural networks[31–35] and using a physics-based model of the stochastic SPD process during training. We call this *physics-aware stochastic training*. While a high SNR for the output of the final layer is likely essential for deterministic inference, our approach allows all the previous layers to be operated in the highly stochastic regime with SNR near 1 (Fig. 1a). This is in contrast with an approach like quantization-aware training[12,36,37], which is only able to deal with quasi-deterministic systems where the SNR in every layer is high.

We experimentally implemented a stochastic ONN using as a building block an optical matrix-vector multiplier (MVM)[12] modified to have SPDs at its output: we call this a *single-photon-detection neural network* (SPDNN). We present results showing that high classification accuracy can be achieved even when the number of photons per neuron activation is ~1, and even without averaging over multiple shots. We also studied in simulation how larger, more

sophisticated stochastic ONNs could be constructed and what their performance on CIFAR-10 image classification would be. In addition to managing extremely low SNR, our probabilistic modeling approach makes SPDNNs inherently robust to various setup imperfections, including dark counts in detectors, errors in optical linear operations, and fluctuations in light intensity (see Supplementary Notes 7, 10, 11). While the proof-of-concept experiments we report are based on a specific spatially multiplexed, free-space ONN, our approach doesn't rely on details of this architecture and could be adapted for many other types of ONN, including those based on diffractive optics[6,11,38], Mach-Zehnder interferometer meshes[5,39,40], and other on-chip or hybrid approaches to matrix-vector multiplication[9,10,15,41].

## Results

### Single-photon-detection neural networks: optical neural networks with stochastic activation from single-photon detection

We consider ONNs in which one or more layers are each constructed from an optical MVM followed by an array of SPDs (Fig. 2a–c), and in which the optical powers used are sufficiently low that in each execution of the layer, each SPD has at most only a few photons impinging on it, leading to stochastic measurement outcomes of *no click* or *click*.

In our setting, we aim to perform *inference* using the SPDNN—with its implementation in physical hardware—(Fig. 2d) and to perform *training* of the SPDNN in silico (Fig. 2e). That is, training is performed entirely using standard digital electronic computing. It is not required that the training be done in silico for it to succeed but is just a choice we made in this work. *Hardware-in-the-loop* training, such as used in ref. 22, is a natural alternative to purely in silico training that even can make training easier by relaxing the requirements on how accurate the in silico model of the physical hardware process needs to be.

**Physics-aware stochastic training.** To train an SPDNN, we perform gradient descent using backpropagation, which involves a forward pass, to compute the current error (or loss) of the network, and a backward pass, which is used to compute the gradient of the loss with respect to the network parameters; our procedure is inspired by backpropagation-based training of stochastic and binary neural networks[31,34]. We model the forward pass (upper part of Fig. 2e) through the network as a stochastic process that captures the key physics of SPD of optical signals having Poissonian photon statistics[28]: the measurement outcome of SPD is a binary random variable (*no click* or *click*) that is drawn from the Bernoulli distribution with a probability that depends on the mean photon number of the light impinging on the detector. However, during the backward pass (lower part of Fig. 2e), we employ a deterministic mean-field estimator to compute the gradients. This approach avoids the stochasticity and binarization of the SPD process, which typically pose difficulties for gradient estimation.

We now give a brief technical description of our forward and backward passes for training; for full details see "Methods" and Supplementary Notes 1A and 2A. We denote the neuron pre-activations of the $l$th stochastic layer of an SPDNN as $\mathbf{z}^{(l)} = W^{(l)}\mathbf{a}^{(l-1)}$, where $\mathbf{a}^{(l-1)}$ is the activation vector from the previous layer ($\mathbf{a}^{(0)}$ denotes the input vector $\mathbf{x}$ of the data to be classified). In the physical realization of an SPDNN, $\mathbf{z}^{(l)}$ is encoded optically (for example, in optical intensity) following an optical matrix-vector multiplier (optical MVM, which computes the product between the matrix $W^{(l)}$ and the vector $\mathbf{a}^{(l-1)}$) but before the light impinges on an array of SPDs. We model the action of an SPD with a stochastic activation function, $f_{SPD}$ (Fig. 2b; Eq. (1)). The stochastic output of the $l$th layer is then $\mathbf{a}^{(l)} = f_{SPD}(\mathbf{z}^{(l)})$.

For an optical signal having mean photon number $\lambda$ and that obeys Poissonian photon statistics, the probability of a *click* event by an SPD is $P_{SPD}(\lambda) = 1 - e^{-\lambda}$ (Fig. 2c). We define the stochastic

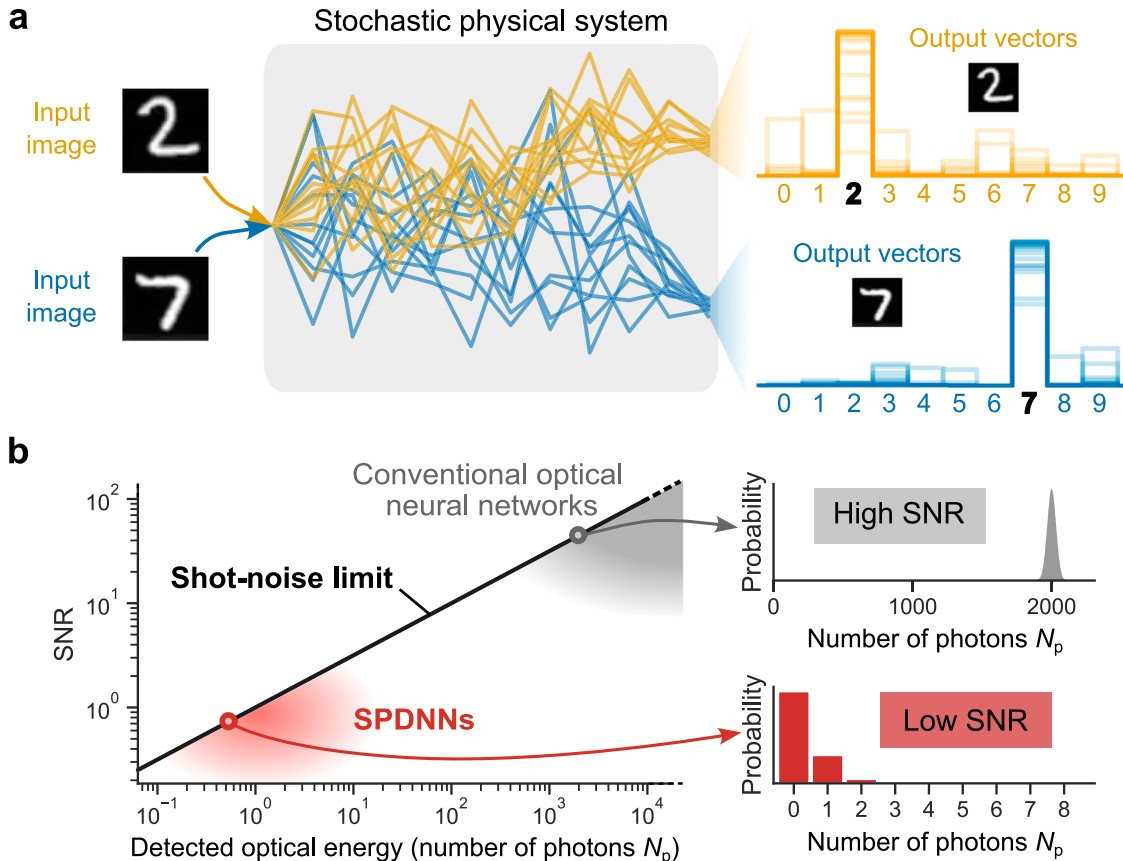

**Fig. 1 | Deterministic inference using noisy neural-network hardware. a** The concept of a stochastic physical neural network performing a classification task. Given a particular input image to classify, repetitions exhibits variation (represented by different traces of the same color), but the class is predicted nearly deterministically. **b** The single-to-noise ratio (SNR) of single-photon-detection neural networks (SPDNNs) compared to conventional optical neural networks (ONNs). Conventional ONNs operate with high photon budgets (SNR ≫ 1) to obtain reliable results, whereas SPDNNs operate with low photon budgets—of up to just a few detected photons per shot (SNR - 1). The relation between the detected optical energy (in number of photons $N_p$) and SNR is SNR $= \sqrt{N_p}$, which is known as the shot-noise limit.

activation function $f_{\text{SPD}}$ as follows:

$$f_{\text{SPD}}(z) := \begin{cases} 1 & \text{with probability } p = P_{\text{SPD}}(\lambda(z)), \\ 0 & \text{with probability } 1-p, \end{cases} \quad (1)$$

where $\lambda(z)$ is a function mapping a single neuron's pre-activation value to a mean photon number. For an incoherent optical setup where the information is directly encoded in intensity, $\lambda(z) = z$; for a coherent optical setup where the information is encoded in field amplitude and the SPD directly measures the intensity, $\lambda(z) = |z|^2$. In general, the form of $\lambda(z)$ is determined by the signal encoding used in the optical MVM, and the detection scheme following the MVM. We use $f_{\text{SPD}}$ in modeling the stochastic behavior of an SPDNN layer in the forward pass. However, during the backward pass, we make a deterministic mean-field approximation of the network: instead of evaluating the stochastic function $f_{\text{SPD}}$, we evaluate $P_{\text{SPD}}(\lambda(z))$ when computing the activations of a layer: $\mathbf{a}^{(l)} = P_{\text{SPD}}(\lambda(\mathbf{z}^{(l)}))$ (Fig. 2b). This is an adaptation of a standard machine-learning method for computing gradients of stochastic neural networks[31].

**Inference.** When performing inference (Fig. 2d), we can run just a single shot of a stochastic layer or we can choose to take the average of multiple shots—trading greater energy and/or time usage for reduced stochasticity. For a single shot, a neuron activation takes on the value $a^{[1]} = a \in \{0, 1\}$; for $K$ shots, $a^{[K]} = \frac{1}{K}\sum_{k=1}^{K} a_k \in \{0, 1/K, 2/K, \dots, 1\}$. In the limit of infinitely many shots, $K \to \infty$, the activation $a^{[\infty]}$ would converge to the expectation value, $a^{[\infty]} = \mathbb{E}[a] = P_{\text{SPD}}(\lambda(z))$. In this work we focus

on the single-shot ($K = 1$) and few-shot ($K \le 5$) regime, since the high-shot ($K \gg 100$) regime is very similar to the high-photon-count-per-shot regime that has already been studied in the ONN literature (e.g., in ref. 12). An important practical point is that averaging for $K > 1$ shots can be achieved by counting the clicks from each SPD. This was done in a clocked fashion by defining a discrete time window during which data was input and the SPDs were monitored for whether they clicked or not (the single-shot, $K = 1$, case); to average for $K > 1$ shots, we kept inputting the data for $K$ times the single-shot discrete time window and summed the number of click events for each SPD. We can think of $K$ as a discrete integration time, so averaging need not involve any data reloading or sophisticated control.

## MNIST handwritten-digit classification with a single-photon-detection multilayer perceptron

We evaluated the performance—both in numerical simulations and in optical experiments—of SPDNNs on the MNIST handwritten-digit-classification benchmark task with a simple, $784 \to N \to 10$ multilayer perceptron (MLP) architecture (Fig. 3a). The activation values in the hidden layer were computed by SPDs. The optical power was chosen so that the SNR of the SPD measurements was -1, falling in the low-SNR regime (Fig. 1b). The output layer was implemented either with full numerical precision on a digital electronic computer, or optically with an integration time set so that the measured signal comprised enough photons that a high SNR (Fig. 1b) was achieved, as in conventional ONNs. Our use of a full-precision output layer is consistent

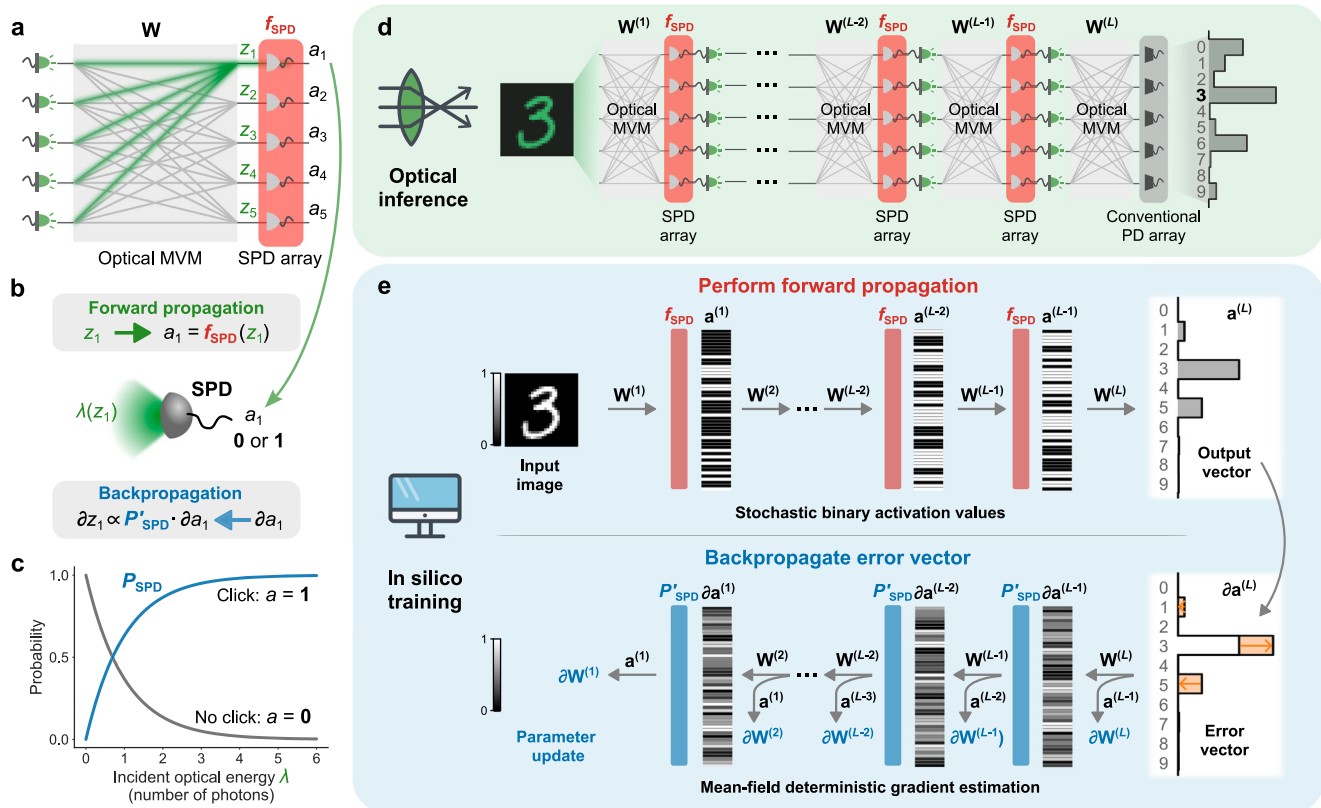

**Fig. 2 | Single-photon-detection neural networks (SPDNNs):** *physics-aware stochastic training* **and** *inference.* **a** A single layer of an SPDNN, comprising an optical matrix-vector multiplier (optical MVM, in gray) and single-photon detectors (SPDs; in red), which perform stochastic nonlinear activations. Each output neuron's value is computed by the physical system as $a_i = f_{SPD}(z_i)$, where $z_i$ is the weighted sum (shown in green) of the input neurons to the $i$th output neuron computed as part of the optical MVM, and $a_i$ is the stochastic binary output from a single-photon detector. **b** Forward and backward propagation through the SPD activation function. The optical energy ($\lambda$) incident on an SPD is a function of $z_i$ that depends on the encoding scheme used. Forward propagation uses the stochastic binary activation function $f_{SPD}$, while backpropagation involves the mean-field function of the probability $P_{SPD}$. **c** Probability of an SPD detecting a click (output $a = 1$) or not (output $a = 0$), as a function of the incident light energy $\lambda$. **d** Optical inference using an SPDNN with $L$ layers. The activation values from the SPD array of each layer are passed to light emitters for the optical MVM of the next layer. The last layer uses a conventional photodetector (PD) array instead of an SPD array. **e** In silico training of an SPDNN with $L$ layers. Each forward propagation is stochastic, and during backpropagation, the error vector is passed to the hidden layers using the mean-field probability function $P_{SPD}$ instead of the stochastic activation function $f_{SPD}$. In this figure, $\partial x$ is shorthand for $\partial C/\partial x$, where $C$ is the cost function.

with other works on binary neural networks[34,42,43]. In a shallow neural network, executing the output layer at high SNR substantially limits the overall energy efficiency gains from using small photon budgets in earlier layers, but in larger models, the relatively high energy cost of a high-SNR output layer is amortized. Nevertheless, as we will see, even with just a single-hidden-layer network, efficiency gains of >40× are possible by performing the hidden layer in the low-SNR regime.

The models we report on in this section used non-negative weights in the hidden layers and real-valued weights in the output layers. This allows the hidden layers to be straightforwardly realized with optical MVMs using incoherent light. A high-SNR layer with real-valued weights can be realized with an incoherent optical MVM if some digital-electronic postprocessing is allowed[12,44]—which is the approach we take for the optical output layer executions in our experiments. However, the postprocessing strategy doesn't directly apply in the low-SNR regime because readout becomes inseparable from the application of a nonlinear activation function, so we are constrained to non-negative weights and activations in the hidden layers. In the next section and Supplementary Note 2, we report on extensions to the case of real-valued weights in coherent optical processors.

**Simulation results.** First, we digitally simulated the SPDNN models shown in Fig. 3a. We report the simulated test accuracies in Fig. 3b for the full test dataset of 10,000 images, as a function of the number of hidden neurons $N$ and the number of shots $K$ of binary SPD measurements integrated to compute each activation.

Due to the stochastic nature of the model, the classification output for a fixed input varies from run to run. We repeated inferences on fixed inputs from the test set 100 times; we report the mean and standard deviation of the test accuracy as data points and error bars, respectively. The standard deviations of the test accuracies are around 0.1%.

If we integrated an infinite number of SPD measurements for each activation ($K \to \infty$)—which is infeasible in experiment, but can be simulated—then the SPDNN output would become deterministic. The test accuracy achieved in this limit can be considered as an upper bound, as the classification accuracy improves monotonically with $K$. Notably, even with just a single SPD measurement ($K = 1$) for each activation, the mean test accuracy is around 97%. The accuracy is substantially improved with just a few more shots of averaging, and approaches the deterministic upper bound when $K \gtrsim 5$. The mean single-photon-detection probability, averaged over all neurons, is ≈0.5, so the simulated number of detected photons per shot is very small: ≈0.5$N$. As we will quantify in the next section reporting the results of optical experiments, this means high accuracy can be achieved using much less optical energy than in conventional ONNs.

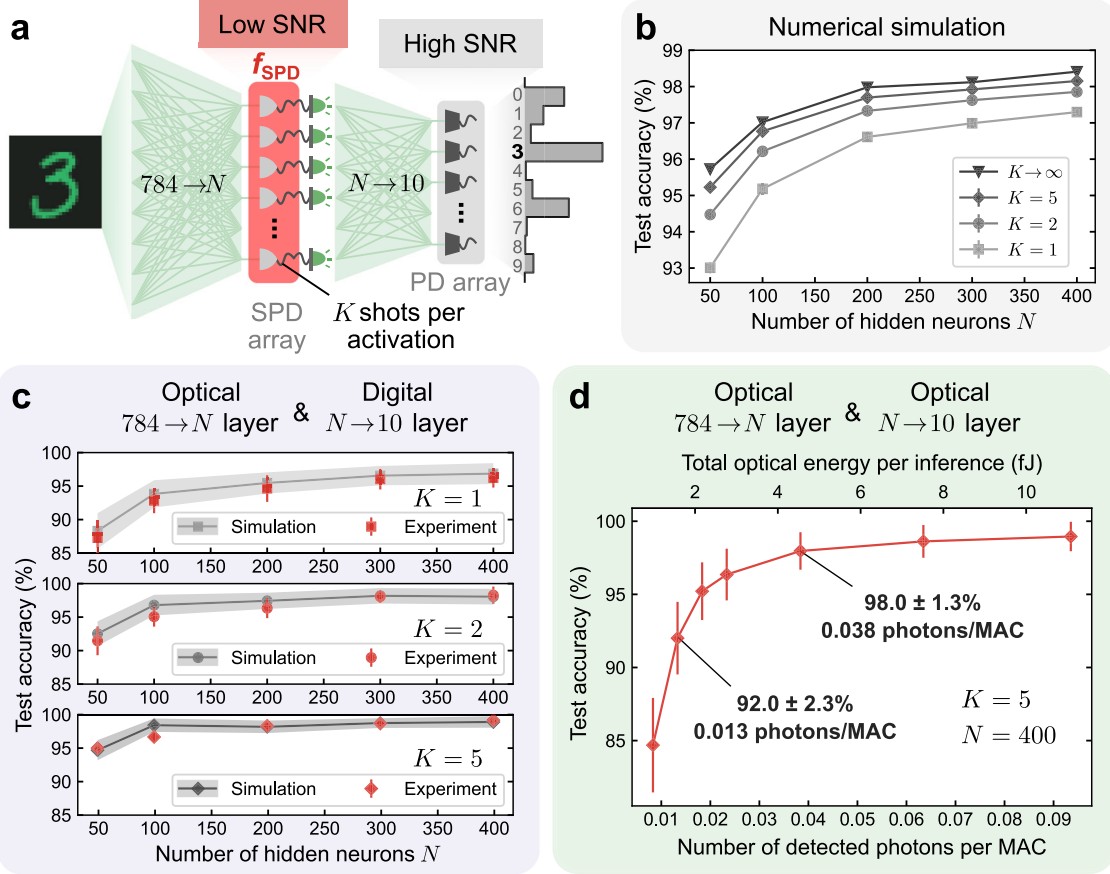

**Fig. 3 | Performance of a single-photon-detection neural network (SPDNN) on MNIST handwritten-digit classification. a** An SPDNN realizing a multilayer perceptron (MLP) architecture of $N$ neurons in the hidden layer. The hidden layer ($784 \to N$) was computed using an incoherent optical matrix-vector-multiplier (MVM) followed by a single-photon-detector (SPD) array. Each SPD realized a stochastic activation function for a single hidden-layer neuron. During a single inference, the hidden layer was executed a small number of times ($1 \leq K \leq 5$), yielding averaged activation values. The output layer ($N \to 10$) was realized either optically—using an optical MVM and high photon budget to achieve high readout SNR, as in conventional ONNs, or with a digital electronic processor, yielding a result with full numerical precision. **b** Simulated test accuracy of MNIST handwritten-digit classification for models with different numbers of hidden neurons $N$ and shots per activation $K$. Error bars, representing standard deviations from 100 repeated

stochastic implementations with identical inputs and weights, are plotted but are too small to be easily visible. **c** Experimental evaluation of the SPDNN, with the output layer performed with full numerical precision on a digital computer. Results are presented for both $K = 1$ (single-shot, i.e., no averaging; top), $K = 2$ (middle), and $K = 5$ (bottom) shots per activation. Mean values and standard deviations (shown as error bars) were calculated from repeated stochastic implementations using identical inputs and weights (see Supplementary Note 8 for details). **d** Experimental evaluation of the SPDNN, with both the hidden and the output layer executed using the optical experimental apparatus. The average number of detected photons used per inference in the hidden layer was kept fixed and the number used per inference in the output layer was varied. Mean values and standard deviations (shown as error bars) were calculated from repeated stochastic implementations using identical inputs and weights (see Supplementary Note 9 for details).

**Optical experimental results.** In our experimental demonstrations, we based our SPDNN on a free-space optical MVM that we had previously constructed for high-SNR experiments[12], and replaced the detectors with SPDs so that we could operate it with ultra-low photon budgets (see "Methods"). The experiments we report were, in part, enabled by the availability of cameras comprising large arrays of pixels capable of detecting single photons with low noise[45]. We encoded neuron values in the intensity of incoherent light, with inputs to each layer encoded using an organic light-emitting diode (OLED) display. As a result, the weights and input-vector elements were constrained to be non-negative for each individual scalar multiplication performed by transmission of light through a single SLM pixel. In the low-SNR layer, this led to the overall matrix-vector multiplication being between non-negative matrices and non-negative vectors. In the high-SNR layer, when performed optically, we used the standard technique for converting an unsigned matrix-vector multiplication into one where the weight matrix is signed[12], which we describe below. However, the restriction to non-negative weights in the low-SNR layer is not a

fundamental feature of SPDNNs—in the next section, we present simulations of coherent implementations that lift this restriction. A single-photon-detecting camera measured the photons transmitted through the optical MVM, producing the stochastic activations as electronic signals that were input to the following neural-network layer (see "Methods", Supplementary Notes 3 and 4).

In our first set of optical experiments, the hidden layer was realized optically, and the output layer was realized in silico (Fig. 3c): the output of the SPD measurements after the optical MVM was passed through a linear classifier executed with full numerical precision on a digital electronic computer. We tested using $K = 1$ (no averaging), $K = 2$, and $K = 5$ shots of averaging the stochastic binary activations in the hidden layer. In Fig. 3c, we can see that the experimental results agree well with results from simulations that additionally modeled the imperfections in our experimental setup (see "Methods", Supplementary Note 7), in contrast to the simulation results shown in Fig. 3b, which did not account for these imperfections. The test accuracies

were calculated using 100 test images, with inference for each image repeated 30 times to enable the computation of error bars. The hidden layer (the one computed optically in these experiments) used ~0.0008 detected photons per MAC ($K = 1$), which is ≥6 orders of magnitude lower than is typical in ONN implementations[10,11,15,26] and ≥3 orders of magnitude lower than the lowest photons-per-MAC numbers reported to date[12,15].

We then performed experiments in which both the hidden layer and the output layer were computed optically (Fig. 3d). In these experiments, we implemented a neural network with 400 hidden neurons and used 5 shots per inference ($N = 400$, $K = 5$). To execute the linear operations with signed weights (matrix elements) on our incoherent setup, we separated the weights into those that were positive and those that were negative. We first performed the matrix-vector multiplication with the negative weights only, then performed the matrix-vector multiplication with the positive weights only, and then obtained the final output by subtracting the results of the first from the second matrix-vector multiplication. The total optical energy was varied by changing the number of photons used in the output layer; the number of photons used in the hidden layer was kept fixed (see "Methods", Table S6 and Supplementary Note 9).

The results show that even though the output layer was operated in the high-SNR regime (Fig. 1b), the full inference computation achieved high accuracy yet used only a few femtojoules of optical energy in total (equivalent to a few thousand photons). By dividing the optical energy by the number of MACs performed in a single inference, we can infer the per-MAC optical energy efficiency achieved: with an average detected optical energy per MAC of ~0.005 attojoules (or 0.014 attojoules) at a photon wavelength of 532 nm, equivalent to 0.013 photons (or 0.038 photons), the mean and standard deviation of test accuracy achieved $92.0 \pm 2.3\%$ (or $98.0 \pm 1.3\%$). The optical energy per MAC is a metric of the energy efficiency of the system while the data is in the optical domain, quantifying a fundamental physical cost of computation, and allows for direct comparison with previous works that report this metric. In practical use of an SPDNN, the end-to-end system energy consumption per inference, covering both optical energy and energy used by the surrounding electronics, would be the preferred metric to optimize for. Engineering end-to-end system advantage is beyond the scope of this paper, but it is an interesting open question to what extent and in which situations reducing the optical energy from a few hundred photons per neuron to just 1 photon per neuron leads to a meaningful decrease in end-to-end system energy usage. Earlier analyses[12,37] in the context of ONNs with few-bit neuron values requiring analog-to-digital and digital-to-analog conversion between layers have concluded that once the optical energy is reduced to be on the order of 1 photon per MAC, the dominant energy cost with currently available electronics is from the electronic components. However, the use of binary stochastic neurons instead of few-bit deterministic neurons involves different hardware and could lead to different conclusions.

We now compare our results with what has been published previously. Our experiments with $N = 50$ hidden neurons and $K = 5$ shots of SPD measurements per activation (see Supplementary Fig. 21) achieved a test accuracy of 90.6% on MNIST handwritten-digit recognition while using only an average of 1390 detected photons per inference (corresponding to ~0.5 fJ of detected optical energy per inference). This represents a >40 × reduction in the number of photons per inference to achieve >90% accuracy on this task versus the previous state-of-the-art[12,15].

While these numbers are already very low, most photons per inference in our experiments were used in the high-SNR output layer, so there is potential for further reduction with a more optimized experimental setup (see Supplementary Note 9). Furthermore, as we discuss in the next section, as the model size is increased, the fraction of optical energy used by a single high-SNR output layer becomes negligible.

## Simulation study of possible future deeper, coherent single-photon-detection neural networks

We have successfully experimentally demonstrated a two-layer SPDNN, but can SPDNNs be used to implement deeper and more sophisticated models? One of the limitations of our experimental apparatus was that it used an intensity encoding with incoherent light and as a result could natively only perform operations with non-negative numbers. In this section, we will show that SPDNNs capable of implementing signed numbers can be used to realize multilayer models (with up to 6 layers), including models with more sophisticated architectures than multilayer perceptrons—such as models with convolutional layers.

ONNs based on coherent light can naturally encode sign information in the phase of the light and have been realized in many different physical platforms, both with fully-connected layers[5,6,46,47] and with convolutional layers[9,10,38,48]. We propose—and study in simulation—SPDNNs using coherent light. Neuron values are encoded in optical amplitudes that are constrained to have phases that are either 0 (positive values) or $\pi$ (negative values). With this encoding, detection by an SPD—which measures intensity and is hence insensitive to phase—results in a stochastic nonlinear activation function that is symmetric about zero (Fig. 4a; see "Methods"). Alternative detection schemes could be employed that would modify the activation function, but we have focused on demonstrating the capabilities of this straightforward case, avoiding introducing additional experimental complexity.

We performed two sets of simulation experiments: one on coherent SPDNNs trained to perform MNIST handwritten-digit classification, and one on coherent SPDNNs trained to perform CIFAR-10 image classification. Figure 4d shows the architectures tested and simulation results for the MNIST benchmark (see "Methods", Supplementary Note 2B). The accuracy achieved by MLPs with either one or two hidden layers was higher than that of the single-hidden-layer MLP simulated for the incoherent case (Fig. 3b), and an architecture with a single convolutional layer followed by two linear layers achieved >99% accuracy even in the single-shot ($K = 1$) regime.

Figure 4e shows the results of simulating variants of a 6-layer convolutional SPDNN (comprising 4 convolutional layers and 2 fully connected, linear layers) on CIFAR-10 image classification. All these simulation results were obtained in the single-shot ($K = 1$) regime. The number of channels in each convolution layer was varied, which affects the total number of MACs used to perform an inference. We observed that the test accuracy increased with the size of the SPDNN, with accuracies approaching those of conventional convolutional neural networks of comparable size[49], as well as of binarized convolutional neural networks[34,50,51]. In the models we simulated that only used SPD as the activation function (i.e., the ones in which there are no "Digital ReLU" blocks), the high-SNR linear output layer had only 4000 MAC operations, so the number of MACs in the high-SNR layer comprises less than 0.01% of the total MACs performed during an inference. The models we simulated are thus sufficiently large that the total optical energy cost would be dominated by the (low-SNR) layers prior to the (high-SNR) output layer. Equivalently, the optical energy cost per MAC would be predominantly determined by the cost of the low-SNR layers. These simulation results illustrate the ability of SPDNNs to scale to larger and deeper models, enabling them to perform more challenging tasks.

## Discussion

Our research is an example of realizing a neural network using a stochastic physical system. Beyond optics, our work is related and complementary to recent investigations in electronic, spintronic, and

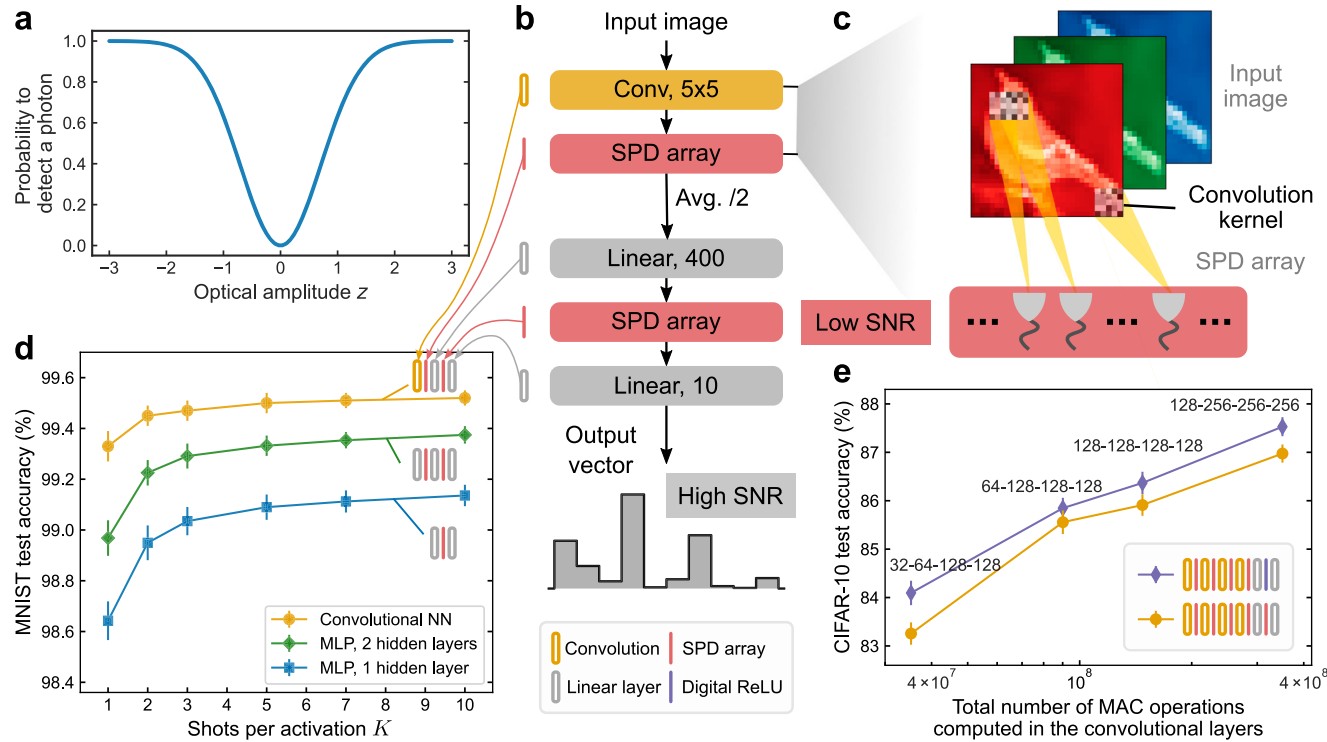

**Fig. 4 | Simulation study predicting the performance of proposed *coherent single-photon-detection neural networks* (SPDNNs). a** The probability of detecting a photon as a function of the input light amplitude in a coherent SPDNN. Real-valued numbers are encoded in coherent light with either 0 phase (positive numbers) or $\pi$ phase (negative numbers). Measurement by a single-photon detector (SPD) results in the probabilistic detection of a photon that is proportional to the square of the encoded value $z$, in comparison to intensity encodings with incoherent light. **b** Structure of a convolutional SPDNN with a kernel size of $5 \times 5$. Single-shot SPD measurements ($K=1$) are performed after each layer (by an SPD array), except for the output layer. Average $2 \times 2$ pooling is applied after each convolutional operation. A digital rectified linear unit (ReLU)[82] activation function can also be used in the linear layer as an alternative. **c** Schematic of a convolutional layer with SPD activations. **d** Simulated test accuracy of coherent SPDNNs with

varying architecture performing MNIST handwritten-digit classification. The multilayer perceptron (MLP) models had 400 neurons in each hidden layer. The convolutional model consisted of a convolutional layer with 16 output channels, followed by two linear layers with an SPD activation inbetween. **e** Simulated test accuracy of coherent SPDNNs with varying architecture performing CIFAR-10 image classification. The models have four convolutional layers, each followed by SPD activation functions. The two linear layers can either be implemented in full-precision with a ReLU activation function (in purple) or using the SPD activation function. The number of output channels for each convolutional layer is indicated above the corresponding data point. The error bars in the plots indicate the standard deviations from 100 repeated stochastic implementations with identical inputs and weights.

---

quantum neuromorphic computing[3,52–59], including in training physical systems to perform neural-network inference[22,60–66].

Neural networks are well-suited for execution on analog devices due to their inherent tolerance to low-precision operations[8]. While previous works (e.g., refs. 37,67,68) have studied energy consumption in analog optical systems, they have focused on the regime where the physical system effectively performs low-precision arithmetic in a way that is quite similar to in digital devices. Our research has instead investigated the regime where the analog system is operating at such a low signal-to-noise ratio (SNR) that it is no longer effective to treat the system simply as a low-precision approximation of a perfect arithmetic engine. Instead, we treat the system as fundamentally stochastic and train neural networks for it by modeling the stochastic physical processes that the system undergoes. Our results show that this *physics-aware stochastic training* can enable ONNs to be operated at substantially lower SNR than what has previously been demonstrated, and in turn have much lower optical energy consumption.

Noise is a fundamental feature of analog machines and ultimately limits the energy efficiency in computing with any analog physical system. It has long been realized that stochasticity is not always detrimental: not only does it not necessarily prevent accurate computation, but can in some cases even enable fundamentally new and more efficient algorithms or types of computation. Our work shows that using a quantum physical model of a particular hardware's inherent

stochastic response at the software level can enable surprisingly large gains in energy efficiency.

While there are many reasons computer science has traditionally favored the abstraction of hardware from software, our work is part of a broad trend, spanning many different physical platforms[4,69,70], in which researchers engineer computations in a physics-aware manner. By short-circuiting the abstraction hierarchy—in our case, going from a physics-aware software description of a stochastic neural network directly to a physical optical realization of the constituent operations— it is possible to achieve orders-of-magnitude improvements in energy efficiency[8,37] versus conventional CMOS computing. *Physics-aware software*, in which software directly incorporates knowledge of the physics of the underlying computing hardware—such as in the *physics-aware stochastic training* we used in this work—is understudied compared to purely software-level or hardware-level innovations (i.e., "at the top" or "at the bottom" of the hierarchy[71]). It is thus ripe for exploration: within the domain of neural networks, there are a multitude of emerging physical platforms that could be more fully harnessed if the physical devices were not forced to conform to the standard abstractions in modern computer architecture[22]. Beyond neural-network accelerators, communities such as computational imaging[72] have embraced the opportunity to improve system performance through co-optimizing hardware and software in a physics-aware manner. We believe there is an opportunity to make gains in

even more areas and applications of computing technology by collapsing abstractions and implementing physics-aware software with physical hardware that could be orders of magnitude faster or more energy efficient than current digital CMOS approaches but that doesn't admit a clean, digital, deterministic abstraction.

## Methods

### Stochastic optical neural networks using single-photon detection as the activation function

In the SPDNNs, the activation function is directly determined by the stochastic physical process of single-photon detection (SPD). Each SPD measurement produces a binary output of either 0 or 1, with probabilities determined by the incident light intensity. Consequently, each SPD neuron activation, which corresponds to an SPD measurement in experiments, is considered as a binary stochastic process[73–75].

Following the Poisson distribution, the probability of an SPD detecting a photon click is given by $P_{SPD}(\lambda) = 1 - e^{-\lambda}$ when exposed to an incident intensity of $\lambda$ photons per detection. Note that these photon statistics may vary based on the state of light (e.g., squeezed light), but here we only consider Poissonian light. Therefore, the SPD process can be viewed as a Bernoulli sampling of that probability, expressed as $f_{SPD}(z) = \mathbf{1}_{t < P_{SPD}(\lambda(z))}$, where $t$ is a uniform random variable $t \sim U[0,1]$ and $\mathbf{1}_x$ is the indicator function that evaluates to 1 if $x$ is true. This derivation leads to Equation (1) in the main text. In our approach, the pre-activation value $z$ is considered as the direct output from an optical MVM that encodes the information of a dot product result. For the $i$th pre-activation value in layer $l$, denoted as $z_i^{(l)}$, the expression is given by:

$$z_i^{(l)} = \sum_{j=1}^{N_{l-1}} w_{ij}^{(l)} \cdot a_j^{(l-1)}, \qquad (2)$$

where $N_{l-1}$ is the number of neurons in layer $l-1$, $w_{ij}^{(l)}$ is the weight between the $i$th neuron in layer $l$ and the $j$th neuron in layer $l-1$, $a_j^{(l-1)}$ is the activation of the $j$th neuron in layer $l$. The intensity $\lambda(z)$ is a function of $z$ that depends on the detection scheme employed in the optical MVM. In optical setups using incoherent light, the information is directly encoded in the intensity, resulting in $\lambda = z$. If coherent light were used in a setup where 0 and $\pi$ phases represent the sign of the amplitude, the intensity would be determined by squaring the real-number amplitude if directly measured, resulting in $\lambda = z^2$. While more sophisticated detection schemes can be designed to modify the function of $\lambda(z)$, we focused on the simplest cases to illustrate the versatility of SPDNNs.

During the inference of a trained model, in order to regulate the level of uncertainty inherent in stochastic neural networks, we can opt to conduct multiple shots of SPD measurements during a single forward propagation. In the case of a $K$-shot inference, each SPD measurement is repeated $K$ times, with the neuron's final activation value $a^{[K]}$ being derived from the average of these $K$ independent stochastic binary values. Consequently, for a single shot, $a^{[1]} = a \in \{0, 1\}$; for $K$ shots, $a^{[K]} = \frac{1}{K} \sum_{k=1}^{K} a_k \in \{0, 1/K, 2/K, \ldots, 1\}$. By utilizing this method, we can mitigate the model's stochasticity, enhancing the precision of output values. Ideally, with an infinite number of shots ($K \to \infty$), the activation $a^{[\infty]}$ would equate to the expected value without any stochasticity, that is, $a^{[\infty]} = \mathbb{E}[a] = P_{SPD}(\lambda(z))$. The detailed process of inference of SPDNNs is described in Algorithm 2 in Supplementary Note 1A.

The training of our stochastic neuron models takes inspiration from recent developments in training stochastic neural networks. We have created an effective estimator for training our SPDNNs while accounting for the stochastic activation determined by the physical SPD process. To train our SPDNNs, we initially adopted the idea of the "straight-through estimator" (STE)[31,76], which enables us to bypass the

stochasticity and discretization during neural network training. However, directly applying STE to bypass the entire SPD process led to subpar training performance. To address this, we adopted a more nuanced approach by breaking down the activation function and treating different parts differently. The SPD process can be conceptually divided into two parts: the deterministic probability function $P_{SPD}$ and the stochasticity introduced by the Bernoulli sampling. For a Bernoulli distribution, the expectation value is equal to the probability, making $P_{SPD}$ the expectation of the activation. Instead of applying the "straight-through" method to the entire process, we chose to bypass only the Bernoulli sampling process. At the same time, we incorporate the gradients induced by the probability function, aligning them with the expectation values of the random variable. In this way, we obtained an unbiased estimator[77] for gradient estimation, thereby enhancing the training of our SPDNNs.

In the backward propagation of the $l$th layer, the gradients of the pre-activation $z^{(l)}$ can be computed as (the gradient with respect to any parameter $x$ is defined as $g_x = \partial C / \partial x$ where $C$ is the cost function):

$$g_{z^{(l)}} = \frac{\partial a^{(l)}}{\partial \lambda^{(l)}} \circ \frac{\partial \lambda^{(l)}}{\partial z^{(l)}} \circ g_{a^{(l)}} = P'_{SPD}(\lambda^{(l)}) \circ \frac{\partial \lambda^{(l)}}{\partial z^{(l)}} \circ g_{a^{(l)}}, \qquad (3)$$

where $a^{(l)} = f_{SPD}(z^{(l)}) = \mathbf{1}_{t < P_{SPD}(\lambda(z^{(l)}))}$ and the gradients $g_{a^{(l)}}$ are calculated from the next layer (previous layer in the backward propagation). Using this equation, we can evaluate the gradients of the weights $W^{(l)}$ as $g_{W^{(l)}} = g_{z^{(l)}}^{\top} a^{(l-1)}$, where $a^{(l-1)}$ are the activation values from the previous layer. By employing this approach, SPDNNs can be effectively trained using gradient-based algorithms (such as SGD[78] or AdamW[79]), regardless of the stochastic nature of the neuron activations.

For detailed training procedures, please refer to Algorithms 1 and 3 in Supplementary Notes 1A and B, respectively.

### Simulation of incoherent SPDNNs for deterministic classification tasks

The benchmark MNIST (Modified National Institute of Standards and Technology database)[80] handwritten digit dataset consists of 60,000 training images and 10,000 testing images. Each image is grayscale with $28 \times 28 = 784$ pixels. To adhere to the non-negative encoding required by incoherent light, the input images are normalized so that pixel values range from 0 to 1.

To assess the performance of the SPD activation function, we investigated the training of the MLP-SPDNN models with the structure of $784 \xrightarrow{W^{(1)}} N \xrightarrow{W^{(2)}} 10$, where $N$ represents the number of neurons in the hidden layer; $W^{(1)}$ ($W^{(2)}$) represents the weight matrices of the hidden (output) layer. The SPD activation function is applied to the $N$ hidden neurons, and the resulting activations are passed to the output layer to generate output vectors (Fig. 3a). To simplify the experimental implementation, biases within the linear operations were disabled, as the precise control of adding or subtracting a few photons poses significant experimental challenges. We observed that this omission has minimal impact on the model's performance.

In addition, after each weight update, we clamped the elements of $W^{(1)}$ in the positive range in order to comply with the constraint of non-negative weights in an incoherent optical setup. Because SPD is not required at the output layer, the constraints on the last layer operation are less stringent. Although our simulations indicate that the final performance is only marginally affected by whether the elements in the last layer are also restricted to be non-negative, we found that utilizing real-valued weights in the output layer provided increased robustness against noise and errors during optical implementation. As a result, we chose to use real-valued weights in $W^{(2)}$.

During the training process, we employed the LogSoftmax function on the output vectors and used cross-entropy loss to formulate the loss function. Gradients were estimated using the unbiased estimator described in the previous section and Algorithm 1.

For model optimization, we found that utilizing the SGD optimizer with small learning rates yields better accuracy compared to other optimizers such as AdamW, albeit at the cost of slower optimization speed. Despite the longer total training time, the SGD optimizer leads to a better model in the end. The models were trained with a batch size of 128, a learning rate of 0.001 for the hidden layer and 0.01 for the output layer, over 10,000 epochs to achieve optimized parameters. To prevent gradient vanishing in the plateau of the probability function $P_{SPD}$, pre-activations were clamped at $\lambda_{max} = 3$ photons.

It should be noted that due to the inherent stochasticity of the neural networks, each forward propagation generates varying output values even with identical weights and inputs. However, we only used one forward propagation in each step. This approach effectively utilized the inherent stochasticity in each forward propagation as an additional source of random search for the optimizer. Given the small learning rate and the significant noise in the model, the number of epochs exceeds what is typically required for conventional neural network training processes. The training was performed on a GPU (Tesla V100-PCIE-32GB) and takes ~8 h for each model.

We trained incoherent SPDNNs with a varying number of hidden neurons $N$ ranging from 10 to 400. The test accuracy of the models improved as the number of hidden neurons increased (see Supplementary Note 1B for more details). During inference, we adjusted the number of shots per SPD activation $K$ to tune the SNR of the activations within the models.

For each model configuration with $N$ hidden neurons and $K$ shots of SPD readouts per activation, we repeated the inference process 100 times to observe the distribution of stochastic output accuracies. Each repetition of inference on the test set, which comprises 10,000 images, yielded a different test accuracy. The mean values and standard deviations of these 100 repetitions of test accuracy are plotted in Fig. 3b (see Supplementary Table 1 for more details). It was observed that increasing either $N$ or $K$ led to higher mean values of test accuracy and reduced standard deviations.

## Experimental implementation of SPDNNs

**Incoherent optical matrix-vector multiplier.** The optical MVM setup utilized in this work is based on the design presented in[12]. The setup comprises an array of light sources, a zoom lens imaging system, a light intensity modulator, and a photon-counting camera. For encoding input vectors, we employed an OLED display from a commercial smartphone (Google Pixel 2016 version). The OLED display features a 1920 × 1080 pixel array, with individually controllable intensity for each pixel. In our experiment, only the green pixels of the display were used, arranged in a square lattice with a pixel pitch of 57.5 μm. To perform intensity modulation as weight multiplication, we combined a reflective liquid-crystal spatial light modulator (SLM, P1920-500-1100-HDMI, Meadowlark Optics) with a half-wave plate (HWP, WPH10ME-532, Thorlabs) and a polarizing beamsplitter (PBS, CCM1-PBS251, Thorlabs). The SLM has a pixel array of dimensions 1920 × 1152, with individually controllable transmission for each pixel measuring 9.2 × 9.2 μm. The OLED display was imaged onto the SLM panel using a zoom lens system (Resolv4K, Navitar). The intensity-modulated light field reflected from the SLM underwent further de-magnification and was focused onto the detector using a telescope formed by the rear adapter of the zoom lens (1-81102, Navitar) and an objective lens (XLFLUOR4x/340, Olympus).

We decompose a matrix-vector multiplication in a batch of vector-vector dot products that are computed optically, either by spatial multiplexing (parallel processing) or temporal multiplexing (sequential processing). To ensure a more accurate experimental implementation, we chose to perform the vector-vector dot products in sequence in most of the data collection. For the computation of an optical vector-vector dot product, the value of each element in either vector is encoded in the intensity of the light emitted by a pixel on the

OLED and the transmission of an SLM pixel. The imaging system aligned each pixel on the OLED display with its corresponding pixel on the SLM, where element-wise multiplication occurred via intensity modulation. The modulated light intensity from pixels in the same vector was then focused on the detector to sum up the element-wise multiplication values, yielding the vector-vector dot product result. Since the light is incoherent, only non-negative values can be allowed in both of the vectors. For more details for the incoherent optical MVM, please refer to Supplementary Note 3. The calibration of the vector-vector dot products on the optical MVM is detailed in Supplementary Note 5.

**Single-photon-detector array.** In this experiment, we used a scientific CMOS camera (Hamamatsu ORCA-Quest qCMOS Camera C15550-20UP)[45] to measure both conventional light intensity measurement and SPD. This camera, with 4096 × 2304 effective pixels of 4.6 × 4.6 μm each, can perform SPD with ultra-low readout noise in its photon-counting mode. This scientific CMOS camera is capable of carrying out the SPD process with ultra-low readout noise. When utilized as an SPD in the photon-counting mode, the camera exhibits an effective photon detection efficiency of 68% and a dark count rate of ~0.01 photoelectrons per second per pixel (Supplementary Note 4). We typically operate with an exposure time in the millisecond range for a single shot of SPD readout. For conventional intensity measurement that integrates higher optical energy for the output layer implementation, we chose another operation mode that used it as a common CMOS camera. Further details on validating the stochastic SPD activation function measured on this camera are available in Supplementary Note 6.

**Experimental implementation of the SPD activations.** We adapted our SPDNNs training methods to conform to the real-world constraints of our setup, ensuring successful experimental implementation (see Supplementary Note 7). First, we conducted the implementation of the hidden layers and collected the SPD activations experimentally by the photon-counting camera as an SPD array. Each SPD realized a stochastic activation function for a single hidden-layer neuron. During a single inference, the hidden layer was executed a small number of times ($1 \leq K \leq 5$), yielding averaged activation values. Then we performed the output layer operations digitally on a computer. This aims to verify the fidelity of collecting SPD activations from experimental setups. Supplementary Fig. 17 provides a visual representation of the distribution of some of the output vectors. For the experiments with 1 shot per activation ($K = 1$), we collected 30 camera frames from the setup for each fixed input image and weight matrix, which are regarded as 30 independent repetitions of inference. They were then used to compute 30 different test accuracies by performing the output linear layer on a digital computer. For the experiments with 2 shots per activation ($K = 2$), we divided the 30 camera frames into 15 groups, with each group containing 2 frames. The average value of the 2 frames within each group serves as the activations, which are used to compute 15 test accuracies. For additional results and details, please refer to Supplementary Note 8.

**Optical implementation of the output linear layer.** Second, to achieve the complete optical implementation of the entire neural networks, we utilized our optical MVM again to carry out the last layer operations. For example, we first focused on the data from the model with 400 hidden neurons and performed 5 shots per inference. In this case, for the 30 binary SPD readouts obtained from 30 frames, we performed an averaging operation on every 5 frames, resulting in 6 independent repetitions of the inference. These activation values were then displayed on the SLM as the input for the last layer implementation. For the 5-shot activations, the possible values included 0, 0.2, 0.4, 0.6, 0.8, and 1. When the linear operations were performed on

a computer with full precision, the mean test accuracy was ~99.17%. To realize the linear operation with real-valued weight elements on our incoherent optical setup, we divided the weight elements into positive and negative parts. Subsequently, we projected these two parts of the weights onto the OLED display separately and performed them as two different operations. The final output value was obtained by subtracting the results of the negative weights from those of the positive weights. This approach requires at least double the photon requirement for the output layer and offers room for optimization to achieve higher energy efficiency. Nevertheless, even with these non-optimized settings, we demonstrated a photon budget that is lower than any other ONN implementations known to us for the same task and accuracy. For additional data and details, please refer to Supplementary Note 9.

### Deeper SPDNNs operating with coherent light
Optical processors with coherent light have the ability to preserve the phase information of light and have the potential to encode complex numbers using arbitrary phase values. In this work, we focused on coherent optical computing utilizing real-number operations. In this approach, positive and negative values are encoded in the light amplitudes corresponding to phase 0 and $\pi$, respectively.

As the intensity of light is the square of the amplitude, direct detection of the light amplitude, where the information is encoded, would involve an additional square operation, i.e., $\lambda(z) = |z|^2$. This leads to a "V-shape" SPD probability function with respect to the pre-activation $z$, as depicted in Fig. 4a. We chose to focus on the most straightforward detection case to avoid any additional changes to the experimental setup. Our objective is to demonstrate the adaptability and scalability of SPDNN models in practical optical implementations without the need for complex modifications to the existing setup.

### Coherent SPDNNs for MNIST classification. MLP-SPDNNs
Classifying MNIST using coherent MLP-SPDNNs was simulated utilizing similar configurations as with incoherent SPDNNs. The only difference was the inclusion of the coherent SPD activation function and the use of real-valued weights. Contrary to the prior scenario with incoherent light, the input values and weights do not need to be non-negative. The models were trained using the SGD optimizer[78] with a learning rate of 0.01 for the hidden layers and 0.001 for the last linear layer, over a period of 10,000 epochs.

**Convolutional SPDNNs** The convolutional SPDNN model used for MNIST digit classification, illustrated in Fig. 4b, consists of a convolutional layer with 16 output channels, a kernel size of $5 \times 5$, a stride size of 1, and padding of 2. The SPD activation function was applied immediately after the convolutional layer, followed by average pooling of $2 \times 2$. The feature map of $14 \times 14 \times 16 = 3136$ was then flattened into a vector of size 3136. After that, the convolutional layers were followed by a linear model of $3136 \to 400 \to 10$, with the SPD activation function applied at each of the 400 neurons in the first linear layer.

The detailed simulation results of the MNIST test accuracies of the coherent SPDNNs can be found in Supplementary Table 2 with varying model structures and shots per activation $K$. For additional information, see Supplementary Note 2B.

### Coherent convolutional SPDNNs for CIFAR-10 classification.
The CIFAR-10 dataset[81] has 60,000 images, each having $3 \times 32 \times 32$ pixels with 3 color channels, that belong to 10 different categories, representing airplanes, automobiles, birds, cats, deer, dogs, frogs, horses, ships, and trucks. The dataset is partitioned into a training set with 50,000 images and a test set with 10,000 images. The pixel values have been normalized using the mean value of (0.4914, 0.4822, 0.4465) and standard deviation of (0.2471, 0.2435, 0.2616) for each of the color channels. To boost performance, data augmentation techniques

including random horizontal flips (50% probability) and random $32 \times 32$ crops (with 4-pixel padding) were implemented during training.

The convolutional SPDNN models for CIFAR-10 classification have deeper structures. Same as the convolutional models trained for MNIST, the convolutional layers use a kernel size of $5 \times 5$, a stride size of 1 and padding of 2. Each convolutional layer is followed by the SPD activation function, average pooling of $2 \times 2$, as well as batch normalization. After $N_{conv}$ convolutional layers ($N_{conv} = 4$ in Fig. 4e) with the number of output channels of the last one to be $N_{chan}^{last}$, the feature map of $(32/2^{N_{conv}})^2 \times N_{chan}^{last}$ is flattened to a vector, followed by two linear layers of $(32/2^{N_{conv}})^2 \times N_{chan}^{last} \to 400 \to 10$. In the first linear layer, either SPD or ReLU[82] activation functions were used for each of the 400 neurons, as depicted in Fig. 4e. We vary the number of convolutional layers and number of output channels of them to change the different model size (Fig. 4e and Supplementary Fig. 5). In these results, we only used a single shot of SPD measurement ($K = 1$) to compute the SPD activations in the models, including the convolutional and linear layers. For additional information, please refer to Supplementary Note 2C.

## Data availability
The data and code needed to reproduce the results presented in this paper are available for download at https://doi.org/10.5281/zenodo.8188270. We have included the raw data resulting from our numerical (simulation) and optical experiments, the code used to process this data, the training datasets and trained-model parameters, as well as examples to demonstrate the operation of our data-collection and data-processing code.

## Code availability
We have also made available a pedagogical code repository, available at https://github.com/mcmahon-lab/Single-Photon-Detection-Neural-Networks, which may be adapted to train models for different stochastic physical hardware setups.

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

## Acknowledgements

We wish to thank NTT Research for their financial and technical support (S.-Y.M., P.L.M., T.W. and L.G.W.). Portions of this work were supported by the National Science Foundation (award no. CCF-1918549; J.L., P.L.M., and T.W.), a Kavli Institute at Cornell instrumentation grant (P.L.M. and T.W.), and a David and Lucile Packard Foundation Fellowship (P.L.M.). P.L.M. acknowledges membership of the CIFAR Quantum Information Science Program as an Azrieli Global Scholar. T.W. acknowledges partial support from an Eric and Wendy Schmidt AI in Science Postdoctoral Fellowship. We acknowledge discussions with M. Anderson, F. J. Chen, R. Hamerly, T. Onodera, S. Prabhu, M. M. Sohoni and R. Yanagimoto. We also acknowledge Z. Eslami, V. Kremenetski, F. Presutti, C. Wan, and F. Wu for suggestions regarding the manuscript.

## Author contributions

S.-Y.M., L.G.W., T.W., and P.L.M. conceived the project. S.-Y.M. and T.W. designed the experiments and built the experimental setup. S.-Y.M. and J.L. performed the neural-network training. S.-Y.M. performed the experiments, the data analysis, and the numerical simulations. All authors contributed to preparing the manuscript. T.W., L.G.W., and P.L.M. supervised the project.

## Competing interests

The authors declare no competing interests.
