## [Transparent Peer Review file · Nature Communications]

Quantum-limited stochastic optical neural networks operating at a few quanta per activation

Corresponding Author: Dr Shi-Yuan Ma

Version 0:

Reviewer comments:

Reviewer #1

(Remarks to the Author)

I thank the authors for addressing my comments, and these of the other reviewers. Going through the reply to the reviewers and the new manuscript, I consider these comments have been accurately addressed and the manuscript is now ready to be accepted as is for this journal.

(Remarks on code availability)

Reviewer #2

(Remarks to the Author)

I would like to thank the authors for addressing my previous concerns. The manuscript now makes an excellent points with regards to the practice implications as well of their work's novelty.

I am happy to recommend publication in Nature Communications.

(Remarks on code availability)

Reviewer #3

(Remarks to the Author)

Thank you for the thorough addressing of my (hopefully constructive) review criticisms, I am very satisfied with the changes

(Remarks on code availability)

Reviewer #4

(Remarks to the Author)

This paper demonstrated a novel computing concept with extremely low photon numbers in the stochastic region using photon number statistics. The results are novel and the experiments are solid. I support its publication in Nature Communications with further delays.

(Remarks on code availability)
